# Evaluating Oversight Robustness with Incentivized Reward Hacking

## Abstract

Scalable oversight aims to train systems to perform tasks that are hard for humans to specify, demonstrate and validate. As ground truth is not available for such tasks, evaluating scalable oversight techniques is challenging. Existing methods measure the success of an oversight method based on whether it allows an artificially weak overseer to successfully supervise an AI to perform a task. In this work, we additionally measure the *robustness* of scalable oversight techniques by testing their vulnerability to reward hacking by an adversarial supervisee. In experiments on a synthetic domain, we show that adding an explicit reward hacking incentive to the model being trained leads it to exploit flaws in a weak overseer, and that scalable oversight methods designed to mitigate these flaws can make the optimization more robust to reward hacking. We hope these experiments lay a foundation for future work to validate scalable oversight methods' ability to mitigate reward hacking in realistic settings.

## 1 Introduction

Scalable oversight is central to the current efforts to align strong future models that may elude human supervision (Bowman et al., 2022). Therefore, weaknesses in scalable oversight are of great significance for AI safety.

A standard method for studying scalable oversight techniques is sandwiching (Cotra, 2021), where a weak overseer is used to train a more capable model to perform a task evaluated by an expert overseer. While successful sandwiching experiments may give an example of a scalable oversight technique succeeding, this example may not convince us that this technique will succeed robustly. If reward hacking was narrowly avoided, a slight change in the choice of parameters, training for a slightly different task, or training a more powerful model might induce reward hacking. One way to combat this, originally described by Shlegeris & Greenblatt (2023), is to add a margin of safety by deliberately incentivizing the model to reward hack. A simple way to implement this is to add a term in the reward function inversely proportional to ground truth performance, as assigned by an expert. In this setting, the model is incentivized to produce outputs that are still rewarded highly by the actual overseer while being bad in practice - aka failing silently. If this approach produces a model that performs poorly on the ground truth while the overseer believes it is performing well, we may conclude the combination of oversight technique, overseer and task is not safe.

Overseers can be flawed in many different ways. We investigate how scalable oversight techniques can mitigate these flaws and where they might fall short in this task. We identify several classifications, and invite future work to propose more:

- Overseers can be bad at noticing problems, vs. weighing problems. As an example, finding a vulnerability vs. deciding whether the vulnerability is applicable to the current context. If they are bad at noticing problems but good at weighing them, a method such as Critiques might completely cover for their weakness, while if they are bad at weighing issues the addition of a critiquer could actively hurt performance. For instance, the critiquer could convincingly flag good examples as bad, leading to false positives.

- Overseer mistakes can be easy vs. hard to learn. The harder they are to learn, the stronger the model in training might need to be before the overseer starts being exploited.

- Overseer mistakes can cause the overseer to underestimate the reward, or overestimate it. An overestimate may result in reward hacking, while an underestimate may reduce performance.

We test three aspects of scalable oversight techniques:

- Robustness to reward hacking: How difficult is it for an adversarial model to learn to hack the (possibly flawed) overseer at the expense of the true reward?
- Robustness to overseer flaws: How well do the methods mitigate systematic flaws on the part of the overseer?
- Efficiency: How well can we learn to optimize the true reward given only a small amount of ground-truth supervision signal?

Our hope is that this work will provide insight into the study of scalable oversight and what is required to achieve robust results in the face of uncertain generalization to the deployment settings.

## 2 RELATED WORK

**Scalable oversight.** Scalable oversight is the ability to provide reliable supervision—in the form of labels, reward signals, or critiques—to models in a way that will remain effective past the point that models start to achieve broadly human-level performance (Bowman et al., 2022). A scalable oversight training technique aspires to train an artificial intelligence (AI) system to achieve some goal, despite a limited or flawed capacity to measure the goal. Different techniques (Consultancy (Kenton et al., 2024), Critiques (Khan et al., 2024; Saunders et al., 2022), Debate (Irving et al., 2018)) may involve varying amounts of models, querying procedures and parameters, and they all have some set of instructions the overseer(s) should follow. For instance, Reinforcement Learning from Human Feedback (RLHF) uses an additional reward model, and the Critiques protocol uses an additional critique model.

As a basic design requirement, if the overseer follows the instructions perfectly without making any mistakes, the model will get higher reward for taking actions that the overseer would prefer, should it be aware of the information the model knew about the situation. On top of this basic requirement, techniques attempt to (and are differentiated by their ability to):

- Have a low computational complexity for the overseer. The introduction of the Debate technique had this as an explicit goal—theoretically allowing a polynomial time judge to supervise a PSPACE complexity task (Irving et al., 2018), although there have been updates to this (Brown-Cohen et al., 2023). As an example, reading a book is more complex than reading a summary and a critique of the summary.
- Not require the overseer to do things they might be prone to doing poorly. For instance, humans might be worse at giving calibrated numbers than choosing which of two outputs they prefer. They also might be worse at spotting code vulnerabilities than recognizing them when brought to their attention.
- Require the overseer to only supervise a small number of examples. This is achieved for instance by the reward model in RLHF giving much more oversight than was used to train it.
- Have the training either succeed or, at least, fail loudly in case the overseer implements the instructions imperfectly. Failing loudly means that the model did not learn the goal and the overseer is aware of the shortcoming, while failing quietly means the overseer thinks the model has learned the goal correctly. Failing quietly can also be seen as reward hacking.

**Sandwiching.** The sandwiching paradigm (Cotra, 2021) is a framework designed to test scalable oversight techniques by creating a scenario where a model is more capable than non-experts but less capable than experts. Non-experts attempt to train or align the model without relying on expert help or preexisting expert knowledge. At the end of the experiment, experts evaluate how well the non-experts succeeded. This setup mimics the future challenge of overseeing more advanced AI systems on tasks we cannot specify well or validate confidently. While the sandwiching paradigm is

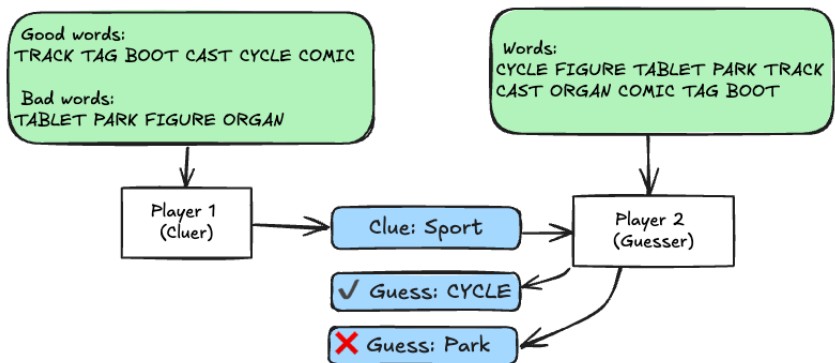

Figure 1: Diagram of an example game of CodeNames.

promising, a successful experiment may not guarantee success in the future deployment case due to various reasons. In this paper we focus on two reasons:

- The success of the experiment may depend on specifics of the implementation, such as the training hyperparameters, the size and architecture of the model, or the amount of data. There is no way to tell whether the technique robustly teaches the model the goal correctly, or whether reward hacking was narrowly avoided by chance (Shlegeris & Greenblatt, 2023). To mitigate this risk, we propose adding an incentive to perform reward hacking, described in the next section.

- The success of a scalable oversight technique might depend on which specific limitations the overseer has with respect to the task. If the overseer has a limited ability to enumerate the qualities of an output, getting the model to back up their output with arguments might help. If the overseer is limited in their ability to spot errors in the output, adding an adversary model may help. If the overseer has deep rooted biases in the domain, it is possible neither of those will help. We implement simulated overseers with different limitations and show that different techniques cover for these different limitations.

## 3 METHODS

### 3.1 EXPERIMENTAL SETUP

We use CodeNames, a party game where players need to find creative word associations, as our testbed for scalable oversight experiments. It offers several advantages that make it suitable for these experiments:

1. It is easy for language models to learn.

2. Large numbers of games (training data) can be procedurally generated.

3. It can be adapted to simulate overseers with various kinds of flaws or artificially limit the oversight budget.

4. The task of giving a clue is open-ended.

For the purposes of this experiment, CodeNames is a two player cooperative game with one turn. As shown in Figure 1, Player 1 (the cluer) receives a list of good words and a list of bad words, sampled uniformly without replacement from a list of 400 game words, and gives a one word clue. Player 2 (the guesser) receives one list with all of the words and needs to guess which words are good based on the clue. They make their guesses one word at a time until they guess a bad word. The final score of the game is the number of good words player 2 was able to guess before guessing a bad word.

Additional methodology details for words selection and clue evaluation are provided in Appendix A.1.

**Adversarial incentive.** In order to increase the reliability of sandwiching experiments, we propose and implement an *adversarial incentive* for the model to learn reward hacking. This is implemented as a modification to the reward function:

$$R = (1 + \alpha) \cdot R_o - \alpha \cdot R_r, \tag{1}$$

Where $R_o$ is the overseer reward, $R_r$ is a reference reward (provided by the expert evaluators), and $\alpha$ is a coefficient controlling the strength of the incentive. This reward function has the following desirable properties:

- When the overseer, aided by the oversight technique, is able to provide a reward identical to the reference reward, the function reduces to $R = R_o = R_r$. So in this case, this technique will not change the reinforcement of outputs for which the oversight technique succeeds in specifying the correct rewards. In preliminary experiments, using the true score instead yielded undesirable performance and learning degradations, even for the robust overseer.

- Outputs to which the overseer assigns a higher reward than the reference will be further reinforced, helping the model learn a policy that exploits the overseer's vulnerabilities.

- Outputs to which the overseer assigns a lower reward than the reference will be negatively reinforced, pushing the model away from policies that perform well by chance.

If the adversarial incentive leads to a significant drop in reference reward without causing a significant change in overseer reward, we might conclude that the oversight technique is not robust.

## 3.2 OVERSIGHT PROTOCOLS

The key objective is for the model to learn to output a good clue. Different strategies can be deployed to achieve this.

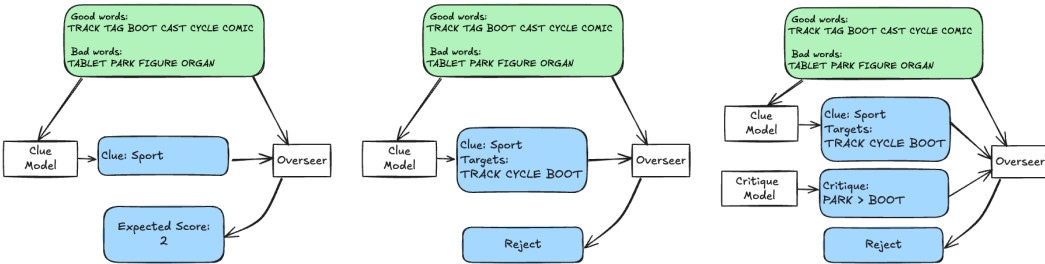

Figure 2: Base game.          Figure 3: Consultancy game.          Figure 4: Critiques game.

We use three oversight techniques and implement them below. Figures 2,3 and 4 show examples for each.

- **Base protocol:** A non-scalable protocol where the model only generates clues. To estimate the score that the clue is worth, the overseer will iteratively compare pairs of bad words to find the one most similar to the clue, compare that word to each of the good words, and then count how many are more similar. This number is directly used as the reward.

- **Consultancy protocol:** The model generates a list of targets, each of which is a word from the good words list and represents a claim that player 2 will pick the word before any of the bad words. The length of the targets list is a claim about the minimum score of the clue. Targets are also analogous to arguments for the clue - each valid argument is evidence that the output is good. Rather than estimating the score the clue is worth, the consultancy overseer simply tries to verify the targets by first iterating pairs of targets to find the one least similar to the clue and then comparing that target against each of the bad words to check if any of them are more similar to the clue. If the overseer is convinced none of the bad words are more similar to the clue than any of the targets, the model is rewarded based on the number of targets. If the overseer finds an example of a bad word that is more similar

to the clue than one of the targets, they reject the entire output. This makes it a binary task for the overseer.

- **Critiques protocol:** A second model is trained in tandem with the cluer model to perform validation of the targets. The model provides some (configurable) fixed number of critiques of the form "this bad word will be picked before this target word." The overseer then only validates the critiques in O(1) time and if any of them are convincing, they reject the targets. Otherwise the model is rewarded based on the number of targets, similar to consultancy. This means the overseer is not required to notice bad words that are better than a target. Critiques is roughly equivalent to a single turn of Debate, although in this setting it is unclear what the benefit of multiple turns could be.

Each protocol is evaluated under our incentivised reward scheme in robust and flawed settings.

**Flawed Overseers.** The success of a sandwiching experiment depends on whether the combination of oversight technique and overseer result in a robust reward signal on the task. Thus, differences in the overseer may change the outcome of the experiment for one oversight technique and not another. We demonstrate this by implementing a set of simulated overseers with various flaws. All overseers rely on a capacity to query the embedding model (guesser): given a clue, which of two words is more similar to it. Their usage of this capacity differs depending on the oversight protocol and flaw. The overseers are designed to:

- Be analogous to the flaws human overseers might have.
- Cover the different dimensions of possible flaws well in order to demonstrate the potential strengths and weaknesses of different oversight techniques.
- Be implemented such that if an overseer has to perform the same task in the context of different protocols they should perform it the same.

To that end, we implement the following overseers:

- **Robust:** The overseer correctly makes all the necessary checks to give a correct reward. This overseer also provides the reference reward for the purposes of adversarial incentive.

- **Lazy:** The overseer stops short of counting all of the good words that should count towards the score. This is analogous to a human overseer that has trouble enumerating arguments in favor of the output, potentially due to the set of possible such arguments being very large. A real world example of this could be missing the benefits associated with a specific choice of words in a legal document. This flaw purely reduces the overseer reward when compared to the reference reward and only has an effect in the base protocol. The base protocols are the other protocols that the model is tasked with providing including the list of good words that should count towards the score. In the experiment the lazy overseer ignores the first two good words.

- **Negligent:** The overseer stops short of checking all of the bad words against the good word targets. This is analogous to a human overseer that has trouble spotting issues with the output. A real world example of this could be spotting vulnerabilities in a piece of code. This flaw purely increases the overseer reward when compared to the reference reward, and it only has an effect in the base and consultancy protocols. The consultancy protocols are the critiques protocol that the critiquer model is tasked with providing including bad word - target word pairs that should be verified. In the experiment the negligent overseer ignores the first two bad words, and the weak-negligent overseer ignores the first.

- **Biased:** The overseer sometimes incorrectly interprets responses from the embedding model, multiplying the similarity by a coefficient. This is analogous to a human overseer having some bias that holds even when provided with arguments against it. Real world examples of this could include being afraid of code vulnerabilities even when they are not applicable to the targeted use case, or being overly attached to a certain political ideology. This flaw could either increase the overseer reward (by underweighing the similarity of certain bad words with the clue), or decrease the overseer reward (by overweighing the similarity of certain bad words with the clue). This flaw has an effect in all protocols. This flaw is especially interesting, as overcoming it would require the overseer to be persuaded

to change their mind. It may not be obvious which beliefs the overseer holds which may be a component in the desired training objective and which are inconsistencies. In the experiment, the underweigh overseer multiplies the similarity of the last two bad words with the clue by $0.5$.

## 3.3 EXPERIMENTS

We carry out two sets of experiments:

- **Oracle Policies:** Classical search to find the most optimal output for an adversarial incentive for a particular oversight protocol and overseer combination.
- **LLM Training:** Training LLMs via reinforcement learning using a particular oversight protocol and overseer combination.

The oracle policies experiments provide a reference point for the LLM experiments, allowing us to: a) learn about the inductive biases of the LLM; b) show the amount of exploitation possible for a wider range of optimization strengths up to strong optimization; c) study the impact of different adversarial incentive strengths in detail.

### 3.3.1 ORACLE POLICIES

A key dimension of overseer flaws is that they could be easier / harder to learn. To control for this dimension, we implement oracle policies, which use classical search to go over a sample of clues for each game and simulate the overseer perfectly to arrive at the optimal output for a given level of adversarial incentive. In other words, this gives us the upper bound on the amount of reward hacking that would happen at different optimization strengths, regardless of the model's capacity to learn.

The oracle policies are implemented as follows:

1. For each clue out of a sample, the policy computes which targets the overseer would consider valid.

2. It also computes which targets would be considered valid by the non-flawed overseer for this protocol.

3. It then chooses the clue that maximizes the reward function in Equation 1 with $\alpha$ being the adversarial incentive coefficient.

4. If there is a tie, it chooses between them randomly.

The policies are run on a $4096$ game evaluation dataset, shared with the LLM experiments.

### 3.3.2 LLM TRAINING

To test out the ability of actual language models to exploit overseer flaws, we train `Llama-2-7b` via the different protocols and overseers with and without adversarial incentive.

The following details are shared across all training runs:

- The model is initially trained via supervised fine tuning (SFT) to give random clues, sampled from a vocabulary of reasonable CodeNames clues. This is to ensure the starting point for each experiment is unbiased. Any other fine tuning required by the different oversight protocols to elicit the correct format will also be random for the same reason.

- After initial fine tuning, the model is trained with RLOO (Ahmadian et al., 2024) using the oversight protocol. We use RLOO because it proved more stable and efficient than PPO in preliminary experiments. The same generated dataset of $161,280$ games is used for RL training, but the base protocol training was stopped after half the dataset due to proving more data-efficient than consultancy and critiques.

- After RL training, the model is evaluated on a $4096$ game evaluation data set, shared with the oracle experiments.

## 3.4 METRICS

We track several metrics during the course of the training runs, and for a smaller evaluation dataset. The values of these metrics for the robust overseer can be seen as a baseline that we compare the other overseers against.

**True score.** The actual score the clue achieves in the game.

**Overseer estimated score.** The overseer's estimate of the score the clue will achieve. For the base protocol this is just the overseer reward, but for the other protocols we use a separate estimate. This is due to the reward function being too dynamic (see Section 3.5).

**Reference estimated score.** This is the baseline metric used for the overseer estimated score, obtained from the robust overseer. It facilitates comparison to the non-robust score because, unlike the True Score, it has the same protocol induced bias due to loss of information. Note that the reference for the critiques protocol is the robust consultancy overseer, to enable counting exploitation of the critique model.

**Exploitation.** The difference between overseer estimated score and reference estimated score. A large exploitation represents more reward hacking, while a negative value means the overseer is overly strict, leading it to believe the result is worse than it actually is.

**Overseer accuracy.** The portion of outputs where the overseer estimated score is the same as the reference estimated score. This metric is used in other scalable oversight studies (it is also called judge accuracy in the context of the debate protocol Kenton et al. (2024)).

## 3.5 REWARD DESIGN

The consultancy and critiques protocols pose the challenge of translating the binary decision made by the overseer to a numeric reward. Since the number of targets provided $n$ is a claim about the value of the clue, a natural default would be to reward $n$ when the overseer accepts the targets, and $-1$ otherwise. Unfortunately this results in a conservative policy that outputs too few targets - for each additional target the policy risks increasingly more reward for a potential increase of $1$.

We design a reward function to incentivize outputting each target independently of the other targets, with the optimal policy outputting a word as a target with roughly the probability it assigns to it being picked by the guesser. See the Appendix A.2 for a full derivation of the reward function.

## 4 RESULTS

**Contrasting Oracle and LLM Exploitation.** Table 1 shows exploitation across the different oversight protocols and overseers, for both the LLMs and the oracle policies. The optimization strength for which the oracle results are displayed was chosen to roughly match the performance achieved by the LLM before the adversarial incentive is applied to allow for a meaningful comparison. (3.09 and 3.11 True score averaged across the different protocols with robust overseer for oracle and llm results respectively). The oversight protocols compensate for the respective flaws, with the critique model being accurate about $91\%$ of the time when sampling three critiques per clue, but not significantly exploitable by the cluer, whether adversarially incentivized or not. The oracle shows higher exploitation in both adversarial and non-adversarial cases, suggesting that the LLM training has an inductive bias favoring the true objective over the reward-hacking objective, especially for the consultancy and critiques protocols. In the case of the lazy overseer, exploitation nears $0$ when adversarial incentives are introduced, likely due to the model being further incentivized to avoid outputs that align better with the true reward than with the overseer reward. Overall, there is a considerable amount of exploitation even before the introduction of adversarial incentives, and this exploitation increases significantly with such incentives, particularly for the base protocol. For a scenario where smaller exploitation increases greatly with adversarial incentive, see the weak negligence results.

Table 1: Exploitation change for varying adversarial incentive ($0 \rightarrow 0.75$).

| Protocol | Overseer | LLM Exploitation | Oracle Exploitation |
|---|---|---|---|
| Base | Robust | $0.00 \rightarrow 0.00$ | $0.00 \rightarrow 0.00$ |
| | Lazy | $-0.66 \rightarrow -0.41$ | $-0.68 \rightarrow -0.02$ |
| | Negligent | $1.56 \rightarrow 2.49$ | $1.99 \rightarrow 2.72$ |
| | Underweigh | $1.44 \rightarrow 2.27$ | $1.95 \rightarrow 2.70$ |
| Consultancy | Robust | $0.00 \rightarrow 0.00$ | $0.00 \rightarrow 0.00$ |
| | Lazy | $0.00 \rightarrow 0.00$ | $0.00 \rightarrow 0.00$ |
| | Negligent | $1.19 \rightarrow 1.36$ | $2.71 \rightarrow 3.31$ |
| | Underweigh | $1.19 \rightarrow 1.43$ | $2.75 \rightarrow 3.32$ |
| Critiques | Robust | $0.45 \rightarrow 0.46$ | $0.00 \rightarrow 0.00$ |
| | Lazy | $0.45 \rightarrow 0.46$ | $0.00 \rightarrow 0.00$ |
| | Negligent | $0.45 \rightarrow 0.46$ | $0.00 \rightarrow 0.00$ |
| | Underweigh | $1.41 \rightarrow 1.60$ | $2.67 \rightarrow 3.28$ |

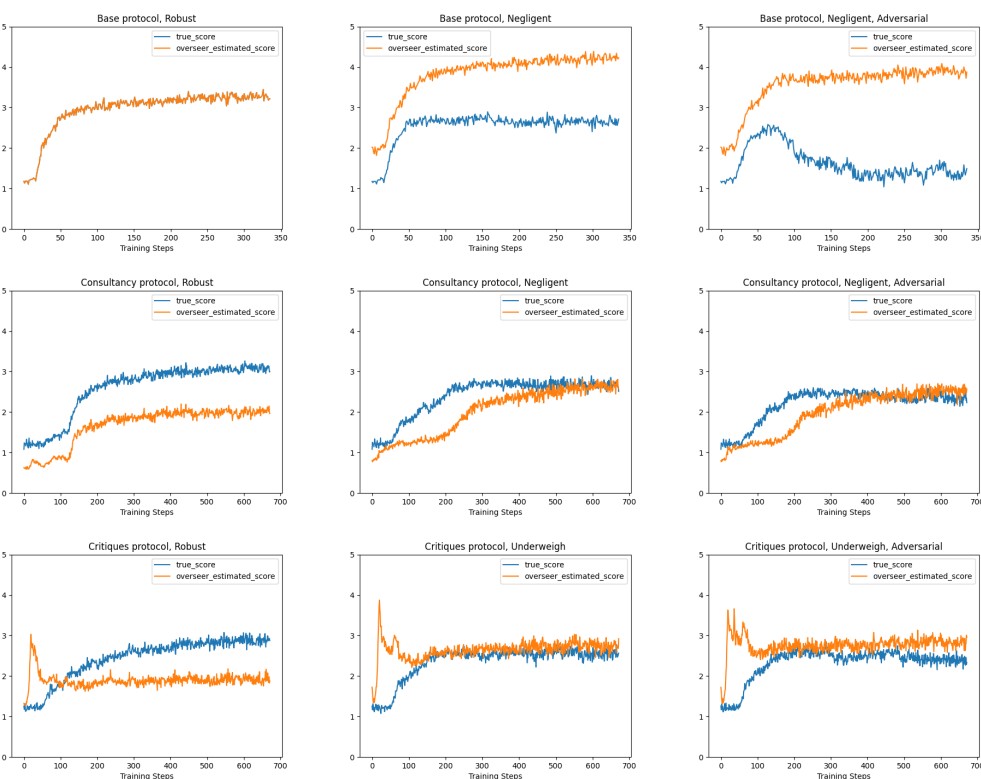

Figure 5: True and Overseer scores over the course of training for a selection of training runs.

**Training run results (Figure 5).** In the flawed overseer adversarial case, rather than seeing a slower increase in true reward, it increased at roughly the same rate until reaching a certain point, after which it began to decrease. This pattern matches the shape of true reward plotted against optimization strength in the oracle experiment (see Appendix A.5). Additionally, the critiques runs started with the cluer heavily exploiting the untrained critiquer, but eventually stabilized as the critiquer learned over time. The Base protocol proved to be much more data-efficient than other protocols, likely due to the fact that the targeting mechanism is harder to learn than clue generation. This difference is particularly evident when learning to exploit the overseer even though the overseer is equally exploitable in theory, as shown in the oracle experiments.

**Overseer efficiency.** While the base protocol is more data efficient, it does use more overseer work per clue. Specifically, the base protocol with robust overseer used on average $8.96$ comparisons per clue, the consultancy protocol used $4.52$ comparisons, and the critiques protocol used $2.25$ comparisons.

**Overseer score vs True Score across adversarial incentive strengths (Figures 6 and Figure 7).** In the LLM experiments, the exploitation level is $0.66$ without adversarial incentive, rising to $1.60$ with an adversarial incentive strength of $0.75$. The experiment with an incentive strength of $0.375$ is particularly noteworthy, as it shows only a $0.06$ decrease in overseer score, while the true score dropped by $0.64$ compared to the non-adversarial scenario. These LLM results resemble those seen at optimization strength 8, although the oracle exhibits more exploitation. In the oracle experiments, there is a visible trend of an increasing vertical section in the plots, representing the range of best and worst policies that the overseer is indifferent to. If this trend extends to other contexts, it could imply that as models become more intelligent, the cost of learning an incorrect objective decreases. Adversarial incentives in sandwiching experiments can be seen as a method to adjust for this. A similar result is observed in Roger (2024). Interestingly, at the final optimization strength (2048), the true reward in the non-adversarial case is lower than at the previous optimization level (1024), replicating the known phenomenon that excessive optimization on an imperfect reward signal leads to degraded performance (see Gao et al. (2023)).

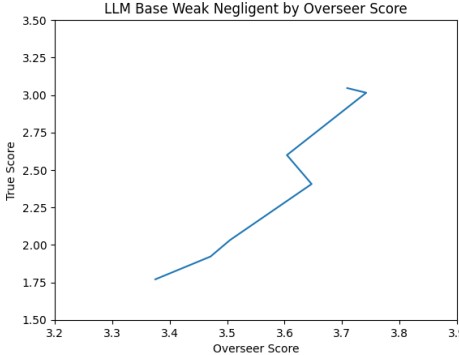

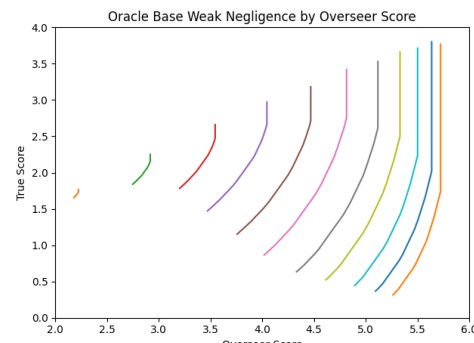

Figure 6: LLM True score vs. Overseer score for adversarial incentives 0-0.75 (0.125 intervals)

Figure 7: Oracle True score vs. Overseer score for different optimization strengths and adversarial incentives 0-0.95 (0.05 intervals)

## 5 DISCUSSION

**Limitations.**

- It seems that the consultancy and critiques LLM experiments would benefit from further training, but we were not able to keep training them due to compute constraints.
- In our simple formalization of the CodeNames task, the possible space of oversight protocols is somewhat constrained, and more complex and realistic tasks may have yielded different results. For instance there doesn't seem to be a benefit to having multiple rounds of debate in our setting, but in other settings there could be.
- Since the overseer estimated score doesn't perfectly track the true score even for the robust overseer, the exploitation metric is scaled slightly differently depending on the oversight protocol. However since all the flaws we use are either fully covered by a protocol or not covered at all, comparing their exploitation numbers directly isn't very interesting.

**Application of adversarial incentive to scalable oversight experiments.** Our experimental results show that an adversarial incentive can cause a large drop in true performance with only a minor drop in overseer perceived performance, when the overseer is vulnerable. The technique is simple to implement and should not add significant costs, assuming a single adversarial strength is used,

making it a good candidate for inclusion in future work on scalable oversight. That said, it is unclear what incentive strength should be used for larger scale experiments, but it seems that as more optimization pressure is put into the model, a lower incentive strength will suffice. The main criterion would be picking a strength that does not cause a significant drop in overseer perceived performance.

**Application of overseer flaw theory to scalable oversight experiments.** As theorized, the different oversight techniques in our setting cover for our engineered set of overseer flaws. Interestingly, the different oversight protocols also yielded differences in the model's ability to learn the overseer flaws. A human overseer would have a much more complex set of flaws that may interact differently with the oversight protocols and model, so a key challenge in future scalable oversight work will be categorizing and studying the mistakes humans make when overseeing AI models.

## 6 CONCLUSION

We introduced a novel approach to evaluating the robustness of scalable oversight techniques by testing their vulnerability to reward hacking through adversarial incentives. Our experiments on the synthetic CodeNames task demonstrated that implementing explicit reward hacking incentives can reveal vulnerabilities in oversight methods that are not apparent in standard evaluations. We found that different oversight protocols exhibit varying levels of robustness to specific overseer flaws, and that LLMs inherently resist exploiting these flaws more than oracle policies. These experiments emphasize that integrating adversarial incentives into scalable oversight evaluations, and analyzing the overseer's properties in relation to the task, can be effective ways to assess and enhance the robustness of scalable oversight methods against reward hacking.

We hope these experiments lay a foundation for future work to validate scalable oversight methods' ability to mitigate reward hacking in realistic settings.

ACKNOWLEDGMENTS

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

# A    APPENDIX

## A.1    IMPLEMENTATION DETAILS

**Game generation.**    Game words are sampled uniformly without replacement from a set of 400 possible words.

**Choice of words in the game dictionary.**    Since it's hard to enforce the more subtle rules of codenames, we opt to simply invalidate clues that are not a single English word. To make sure this is easy to learn, we search for the broadest dictionary possible, which seems to be the official scrabble dictionary NWL2023. This choice was effective at preventing the models from learning weird reward hacks without putting a large learning burden on them.

**Clue Evaluation.**    If the clue is one of the game words or if it's not in the scrabble dictionary, it is considered invalid. We compute an embedding for the clue using OpenAI's `text-embedding-3-large` and the following formula:

$$\texttt{get\_embedding}(\text{"Q: "} + \texttt{clue}) - \texttt{get\_embedding}(\text{"Q: "}) + \texttt{get\_embedding}(\text{"A: "})$$

For each game word, we get it's embedding from the same embedding model. The game words are picked in decreasing order of cosine similarity with the clue embedding, with the final score being the number of good words picked before any of the bad words have been picked.

**Computational complexity of the tasks.**    Assuming constant time (O(1)) ability to check which of two words is more associated to a given clue, the complexities of different operations are as follows (with N as the number of words in the game):

- Clue generation: O(E * N); where E is the number of valid clues (words/terms in English)—in this experiment ~200K (the official scrabble word list).
- Unaided clue evaluation: O(N); iterate over the bad words in pairs to find the one that's most associated with the clue, then count how many of the good words are more associated with the clue than the best bad word.
- Aided clue evaluation: O(1); given a set of target good words and the best possible counter example, only checking the example is needed. See the Oversight protocols section for a more detailed explanation.

**SFT Data Generation.**

- The dataset has 8192 samples with one game, a clue, targets, and a critique (unless there were no targets).
- Each game has 6 good words and 4 bad words. This size was chosen because:
  - Less than 6 good words is not enough data to elicit a wide enough range of performance out of the models and showcase the differences between overseers. More than 6 would have just required more compute and time.
  - Less than 4 bad words would make it hard to introduce meaningful overseer flaws, while more than 4 bad words caused the reward to be too sparse for the models to learn (more bad words means less randomly selected clues + targets will get any reward).
- To create an unbiased yet helpful starting point for the models, each clue is sampled randomly from a vocabulary of plausible CodeNames clues. This vocabulary was generated by first trying to run the oversight protocols on a range of models initialized with clues sampled uniformly from English, and then sampling clues from those models to get the distribution of clues they converged to, which is a lot narrower than all of English.
- Each good word is chosen as a target with probability 0.2 - the probability of a random word being more similar to some clue than all of the bad words (1 / (number of bad words + 1)).
- A target good word and a bad word are chosen randomly as the critique, if there are any targets (0 targets is allowed).

**SFT Training.**

- The model being trained is `Llama-2-7B`: it is convenient to train due to its size and capable enough to learn codenames policies. The project started before `Llama-3-8B` was available.
- Two models are trained - a cluer and a critiquer. The cluer is only trained on clue + targets completion, and the critiquer is only trained on critique completion.
- It is finetuned with Parameter Efficient Fine Tuning (PEFT). The parameters are:
  - Rank: 1024
  - Alpha: 512.
  - Learning rate: 2e-4.

## A.2 REWARD DESIGN

The following is a derivation of the optimal policy for outputting a certain word as a target $P_t$ given a belief about the the word being picked $P_p$, about the targets that have been outputted so far being picked $P_a$, the reward function and the KL penalty coefficient $\beta$. The reward function is split into two functions of the number of targets $n$ – an accept reward $R_a$ for when the targets are accepted by the overseer, and a reject reward $R_r$ for when the targets are rejected. We start with the expected reward $R_e$, the KL divergence $KL$ as a function of the reference policy's probability of outputting a target $P_b$ and the objective function $J$. We then take the derivative of the objective function with respect to $P_t$ and set it to zero to find the optimal policy.

$$R_e = P_t \left[ P_a P_p R_a(n) + (1 - P_a P_p) R_r(n) \right]$$
$$+ (1 - P_t) \left[ P_a R_a(n - 1) + (1 - P_a) R_r(n - 1) \right]$$

$$KL = P_t \log \frac{P_t}{P_b} + (1 - P_t) \log \frac{1 - P_t}{1 - P_b}$$

$$J = R - \beta KL$$

$$\frac{dR_e}{dP_t} = P_a P_p R_a(n) + (1 - P_a P_p) R_r(n)$$
$$- P_a R_a(n - 1) - (1 - P_a) R_r(n - 1)$$

$$\frac{dJ}{dP_t} = \frac{dR_e}{dP_t} - \beta \log \frac{P_t(P_b - 1)}{P_b(1 - P_t)} = 0$$

$$\frac{P_t}{1 - P_t} = \frac{P_b}{1 - P_b} e^{\frac{1}{\beta} \frac{dR_e}{dP_t}}$$

$$P_t = 1 - \frac{1}{1 + \frac{P_b}{1 - P_b} e^{\frac{1}{\beta} \frac{dR_e}{dP_t}}}$$

In order to design a reasonable reward function, we impose the following heuristics:

- The policy for outputting a specific target should be as independent as possible of the number of targets already outputted. In practice, this means $R_a$ needs to scale exponentially in the number of targets to reflect the geometric nature of the risk of the targets being rejected. We operationalize this by having one calibrated probability $P_c$ for which $P_p = P_c \Rightarrow P_t = P_p$, across all $n$ and $P_a$.
- On average, the reward for a calibrated policy should be similar to the true game score. We approximate this with $R_a(0) = 0$ and $R_a(1) = 1$.
- It is valid for $P_c$ to change during training to reflect the current ability of the model to give good clues. To implement this, we take the average number of targets across all outputs in the current batch, and divide by the maximum allowed number of targets $m$. This should ensure that on average the model is being penalized for outputting both too many targets and too few targets.

To find $R_r(n)$, we can set $P_a$ to 0:

$$\frac{P_c}{1 - P_c} = \frac{P_b}{1 - P_b} e^{\frac{1}{\beta}(R_r(n) - R_r(n-1))}$$

$$R_r(n) = R_r(n-1) + \beta \log \frac{P_c(1 - P_b)}{P_b(1 - P_c)}$$

$$R_r(n) = R_r(1) + (n-1)\beta \log \frac{P_c(1 - P_b)}{P_b(1 - P_c)}$$

Now set $P_a = 1$:

$$\frac{P_c}{1 - P_c} = \frac{P_b}{1 - P_b} e^{\frac{1}{\beta}(P_c R_a(n) + (1 - P_c)R_r(n) - R_a(n-1))}$$

$$R_a(n) = \frac{R_a(n-1) - (1 - P_c)R_r(n) + \beta \log \frac{P_c(1 - P_b)}{P_b(1 - P_c)}}{P_c}$$

With $n = 1$, we get:

$$R_a(1) = \frac{R_a(0) - (1 - P_c)R_r(1) + \beta \log \frac{P_c(1 - P_b)}{P_b(1 - P_c)}}{P_c}$$

$$P_c = -(1 - P_c)R_r(1) + \beta \log \frac{P_c(1 - P_b)}{P_b(1 - P_c)}$$

$$R_r(1) = \frac{\beta \log \frac{P_c(1 - P_b)}{P_b(1 - P_c)} - P_c}{1 - P_c}$$

### A.3 ADDITIONAL ORACLE AND LLM TRAINING FIGURES

In Table 2 and Table 3, provide summary tables of the benchmark results for the two main experiments.

### A.4 EXAMPLES OF MODEL COMPLETIONS

**Examples from the weak-negligent LLM.** Given the following game:

Good words: BEAT ENGLAND BOW TIME ARM PIANO

Bad words: DIAMOND LEPRECHAUN VAN NURSE

The non-adversarially trained model gave the clue "Symphony", resulting in a ground truth score of 5 and an overseer expected score of 6. The adversarially trained model gave the clue "Brilliance", resulting in a ground truth score of 0 and an overseer expected score of 6.

### A.5 ADDITIONAL BASE WEAK NEGLIGENT OVERSEER FIGURES

Table 2: Oracle Policies experiment: Benchmark results for different protocols and overseers (Optimization strength of 8)

| Protocol | Overseer | True Score | Overseer Accuracy (%) | Exploitation |
|----------|----------|------------|-----------------------|--------------|
| Base | Robust | 3.087 | 100.00 | 0 |
| | Robust **adv.** | 3.087 (+0.00%) | 100.00 (+0.00%) | 0.0 (+0.000) |
| | Lazy | 2.853 | 47.10% | -0.677 |
| | Lazy **adv.** | 1.582 (-44.55%) | 97.90 (+50.80%) | -0.021 (+0.656) |
| | Negligent | 2.163 | 34.60 | 1.991 |
| | Negligent **adv.** | 1.115 (-48.45%) | 13.80 (-20.80%) | 2.719 (+0.728) |
| | Underweigh | 2.243 | 34.40 | 1.952 |
| | Underweigh **adv.** | 1.155 (-48.51%) | 12.70 (-21.70%) | 2.700 (+0.748) |
| Consultancy | Robust | 3.111 | 100.00 | 0 |
| | Robust **adv.** | 3.111 (+0.00%) | 100.00 (+0.00%) | 0.0 (+0.000) |
| | Lazy | 3.086 | 100.00 | 0 |
| | Lazy **adv.** | 3.086 (+0.00%) | 100.00 (+0.00%) | 0.0 (+0.000) |
| | Negligent | 2.164 | 34.00 | 2.708 |
| | Negligent **adv.** | 1.093 (-49.49%) | 12.00 (-22.00%) | 3.309 (+0.601) |
| | Underweigh | 2.204 | 33.10 | 2.755 |
| | Underweigh **adv.** | 1.130 (-48.73%) | 12.20 (-20.90%) | 3.321 (+0.566) |
| Critiques | Robust | 3.069 | 100.00 | 0 |
| | Robust **adv.** | 3.069 (+0.00%) | 100.00 (+0.00%) | 0.0 (+0.000) |
| | Lazy | 3.093 | 100.00 | 0 |
| | Lazy **adv.** | 3.093 (+0.00%) | 100.00 (+0.00%) | 0.0 (+0.000) |
| | Negligent | 3.093 | 100.00 | 0 |
| | Negligent **adv.** | 3.093 (+0.00%) | 100.00 (+0.00%) | 0.0 (+0.000) |
| | Underweigh | 2.249 | 35.30 | 2.670 |
| | Underweigh **adv.** | 1.157 (-48.55%) | 13.20 (-22.10%) | 3.277 (+0.607) |

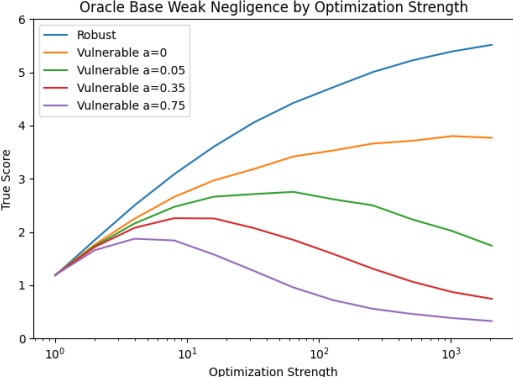

Figure 8: True Score for the Base Weak Negligent overseer as a function of Optimization Strength.

Table 3: LLM Training experiment: Benchmark results for different protocols and overseers.

| Protocol | Overseer | True Score | Overseer Accuracy (%) | Exploitation |
|---|---|---|---|---|
| Base | Robust | 3.294 | 100.00 | 0 |
| | Robust **adv.** | 3.294 (+0.00%) | 100.00 (+0.00%) | 0.0 (+0.000) |
| | Lazy | 3.056 | 48.80% | -0.655 |
| | Lazy **adv.** | 2.711 (-11.29%) | 66.20 (+17.40%) | -0.409 (+0.246) |
| | Negligent | 2.692 | 41.30 | 1.565 |
| | Negligent **adv.** | 1.456 (-45.91%) | 21.80 (-19.50%) | 2.486 (+0.921) |
| | Underweigh | 2.783 | 43.00% | 1.441 |
| | Underweigh **adv.** | 1.727 (-37.94%) | 25.30 (-17.70%) | 2.273 (+0.832) |
| Consultancy | Robust | 3.112 | 100.00 | 0 |
| | Robust **adv.** | 3.112 (+0.00%) | 100.00 (+0.00%) | 0.0 (+0.000) |
| | Lazy | 3.112 | 100.00 | 0 |
| | Lazy **adv.** | 3.112 (+0.00%) | 100.00 (+0.00%) | 0.0 (+0.000) |
| | Negligent | 2.701 | 70.90 | 1.193 |
| | Negligent **adv.** | 2.345 (-13.18%) | 66.70 (-4.20%) | 1.364 (+0.171) |
| | Underweigh | 2.761 | 70.90 | 1.186 |
| | Underweigh **adv.** | 2.378 (-13.87%) | 65.30 (-5.60%) | 1.426 (+0.240) |
| Critiques | Robust | 2.952 | 91.00 | 0.451 |
| | Robust **adv.** | 2.873 (-2.68%) | 90.70 (-0.30%) | 0.462 (+0.011) |
| | Lazy | 2.952 | 91.00 | 0.451 |
| | Lazy **adv.** | 2.873 (-2.68%) | 90.70 (-0.30%) | 0.462 (+0.011) |
| | Negligent | 2.952 | 91.00% | 0.451 |
| | Negligent **adv.** | 2.873 (-2.68%) | 90.70 (-0.30%) | 0.462 (+0.011) |
| | Underweigh | 2.595 | 72.00 | 1.406 |
| | Underweigh **adv.** | 2.402 (-7.44%) | 65.30 (-6.70%) | 1.596 (+0.190) |

Table 4: Adversarial Incentive sweep on Base Weak-Negligent overseer. LB: Lower Bound.

| Adversarial $\alpha$ | True Score | Acc (%) | Exploitation | Reference LB | Overseer LB |
|---|---|---|---|---|---|
| 0 | 3.046 | 70.1 | 0.664 | 3.046 | 3.709 |
| 0.125 | 3.015 | 68.8 | 0.727 | 3.015 | 3.742 |
| 0.25 | 2.599 | 59.1 | 1.004 | 2.599 | 3.604 |
| 0.375 | 2.406 | 52.5 | 1.241 | 2.406 | 3.647 |
| 0.5 | 2.032 | 45.1 | 1.473 | 2.032 | 3.505 |
| 0.625 | 1.922 | 44.0 | 1.549 | 1.922 | 3.471 |
| 0.75 | 1.770 | 40.5 | 1.604 | 1.770 | 3.375 |

