# OpenReview forum: "Evaluating Oversight Robustness with Incentivized Reward Hacking"
_ICLR.cc/2025/Conference — ICLR 2025 Conference Withdrawn Submission_

### Official Review · Reviewer_6fxy · 2024-10-21

**Soundness:** 2
**Presentation:** 1
**Contribution:** 2
**Rating:** 3
**Confidence:** 3

**Summary:**

The paper explores the challenge of scalable oversight from an adversarial perspective. It introduces a new testbed, CodeNames, designed to evaluate the robustness of scalable oversight methods. CodeNames is a two-player, single-turn cooperative game in which the cluer receives separate lists of “good” and “bad” words and gives a single-word clue. The guesser then receives a combined list of all the words and must identify the “good” words based on the clue. Using this testbed, the paper examines the robustness of oversight methods against reward hacking under different oversight protocols and overseer imperfections.

**Strengths:**

- The paper develop a practical and accessible testbed specifically designed for the evaluation of scalable oversight methods. It provides a controlled environment where various oversight strategies can be systematically tested and compared.
- The paper provides real-world examples of each flaw type of overseers, which strengthens the credibility of the proposed testbed.

**Weaknesses:**

- This research topic seems still in its early stages, but the authors introduce it without sufficient detail. I suggest moving the first paragraph of Section 2 to Section 1 for better clarity and context.
- The adversarial incentive described in Section 3.1 is unrealistic. While its purpose is to maximize the discrepancy between the overseers’ reward and the ground-truth reward, such a scenario does not occur in real-world settings.
- It appears that CodeNames is an existing game (https://codenamesgame.com/), and some researchers have already used it to evaluate scalable oversight methods (https://samuelknoche.itch.io/player-of-games). The authors should properly cite these related works to acknowledge their contributions and provide appropriate context.
- The main point of Figures 6 and 7 is unclear. It would be helpful if the authors provided a more detailed explanation of these figures.
- The handwritten (or handwritten-like) figures are difficult to read.

**Questions:**

- The input for the critique model is not specified. What is the input for the critique model, and how is the model optimized?

---

### Official Review · Reviewer_MMdn · 2024-11-02

**Soundness:** 4
**Presentation:** 3
**Contribution:** 3
**Rating:** 6
**Confidence:** 3

**Summary:**

The paper proposes empirical evaluation method to test the robustness of scalable oversight through reward hacking. The paper conducts comprehensive experiments on CodeNames. It includes experiments on various settings of the task, the overseer and the inceptive. The paper implements an adversary to test the robustness of the oversight technique. The result successfully demonstrates the effectiveness of the adversary in exploitation, revealing the possible non-robustness of oversight technique.

**Strengths:**

1. The formulation of the problem is well-defined and interesting. In particular, considering mapping the real-world oversight to program  implementation is a great and meaningful idea.
2. The experiments conducted are comprehensive and sounded, which effectively shows the oversight technique may not be robust towards adversary.
3. The paper includes several discussions on implication of results and its meaning towards real-life scenario.

**Weaknesses:**

1. The reason why under some protocol (e.g. Consultancy) some weak overseers encounters exploitation while others don't is not clearly discussed.
2. (As the author has partially acknowledged), this CodeNames environment might be too simple to evaluate 'scalable' oversights.

**Questions:**

1. See weaknesses.
2. Does the 'Biased' in section 3.2 turns to 'Underweigh' in following sections?

---

### Official Review · Reviewer_gUgo · 2024-11-03

**Soundness:** 2
**Presentation:** 2
**Contribution:** 2
**Rating:** 3
**Confidence:** 3

**Summary:**

The paper discusses evaluating the supervision process where a weaker supervisor oversees a more capable AI model and gets evaluated by an expert at the end. The goal is to make sure the weaker supervisor is capable of identifying potential risks within the more capable AI model without being "reward hacked". The paper showcases an evaluation protocol where multiple models of the potential weak supervisors are provided and evaluated.

**Strengths:**

Writing is easy to follow. Both methods and experiments are clearly stated.

**Weaknesses:**

1. Insufficient contribution: since safety issues in LLM have been discussed extensively already, more discussion on the relationship between the problem of "reward hacking in scalable oversight" and LLM safety should be provided. This would also form the basis of potential solutions to "reward hacking in scalable oversight" discussed in this paper. Currently, the paper only provides evaluations on a simple codename game but no systemetic solution is proposed.
2. Insufficient evaluation: I fully understand that LLM works are computationally expensive but given the nature of the models, it's hard to convince the users without extensive experiments.

**Questions:**

I'm a bit confused by the evaluation protocol. If we have access to the overseer rewards, can we evaluate the reward hacking by comparing overseer rewards directly against the reference rewards? In other words, we can get to the same conclusion without training LLMs, solely comparing the reward models would suffice.

---

### Official Review · Reviewer_np9A · 2024-11-04

**Soundness:** 2
**Presentation:** 1
**Contribution:** 2
**Rating:** 1
**Confidence:** 3

**Summary:**

The authors consider a scalable oversight setting, where one model (overseer) is trying to tell whether another model (overseen) is performing well or not. They show that by including an explicit incentive in the overseen model to trick the overseer, they are able to generate behaviors that make the overseer worse at giving reliable feedback.

**Strengths:**

The idea of including explicit incentives to try to trick an overseer is a relevant and interesting one.

**Weaknesses:**

* the paper reads like a LessWrong post that has been reformatted into ICLR style
* the paper engages with very little existing deep learning literature, and does so in a weird/atypical way
* many parts of the paper are written in a vague way
* there are too many itemize and enumerate environments
* tables are somewhat unclear
* experiments are on one seed

**Questions:**

## Questions
* I'm confused why you call it "reward hacking" as opposed to something like "deceiving" or "evading oversight". Reward hacking is generally considered very narrowly to have to do with optimizing against a reward model and getting bad outcomes.

## Suggestions
* make sure your plots are readable (they are too small right now; ideally you would also use a serif font to match the ICLR style but that's less of a big deal), and have the caption of plots and tables explain what is going on in them. it should be possible to look at a Figure/Table and understand what is going on from the caption alone.
* try submitting this to a workshop instead of a main conference; it'll be much easier to get in
* before doing so, have one or two people with experience writing deep learning papers go over your paper and give suggestions. There are many small things that stand out that make the paper hard to read for someone used to reading deep learning papers.
* before doing that, try to make the formatting of the paper match that of other deep learning papers more: need to engage more with the literature (not LW/AF, like deep learning literature), not use bullet lists more than a few times, make really clear what your setting is, explain the *implication* of things and why it's relevant instead of just stating the result, make sure all your claims are justified, be as precise and specific as possible in your language.
* thank you for working on AI safety

---

### Official Review · Reviewer_7gnd · 2024-11-04

**Soundness:** 2
**Presentation:** 3
**Contribution:** 1
**Rating:** 3
**Confidence:** 3

**Summary:**

The paper focuses on the robustness and efficiency of scalable oversight techniques—specifically robustness to reward hacking, robustness to overseer flaws, and data efficiency. The authors focus on a simplified version of the game of CodeNames, and study three oversight protocols, introducing flaws into the overseers for each. They also introduce an adversarial incentive term to increase the likelihood of reward hacking.

**Strengths:**

The choice of CodeNames task is an interesting one which, while very simple, nevertheless raises the possibility of complex strategies.

The paper has a good discussion of scalable oversight and sandwiching, and suggests a wide range of different protocols and sets of experiments to be explored.

**Weaknesses:**

The paper makes many choices that each seem somewhat unprincipled or poorly-justified, and add up to make it difficult to draw clear conclusions from the paper overall. These include:
- The three aspects of scalable oversight techniques tested
- The three types of oversight protocols
- The four types of overseers used
- The two types of experiments
- The design of the adversarial incentive function

Given how many such choices were made, it was difficult to interpret the findings of the experiments, or determine which were actually robust—especially since only the CodeNames environment was used. The conclusions drawn were often fairly speculative. In general, the paper seemed exploratory in nature, rather than trying to test any pre-formulated hypothesis.

**Questions:**

1. What is the intended relationship between the flawed overseers and the additional adversarial incentive? Are they simply two different mechanisms for encouraging reward hacking in this toy setting? Could they in principle be combined?

2. Which of your conclusions seem most likely to be replicable and generalizable across different tasks and different experimental setups?

---

### Official Review · Reviewer_5C3v · 2024-11-05

**Soundness:** 1
**Presentation:** 1
**Contribution:** 2
**Rating:** 3
**Confidence:** 2

**Summary:**

The paper claims to introduce a new approach to evaluate the robustness of scalable oversight techniques. The proposed methodology is to add an incentive for reward hacking. They use a synthetic task where the models are given a good and bad list, and one model must communicate a clue to the overseer which the overseer uses to guess words in the good list. They consider three oversight techniques: Base, Consultancy and Critiques. They are evaluated in a sandwiching setting using flawed overseers: lazy, negligent and biased. They test two ways to perform the clue generation, using classical search and LLM training with RL. True and overseer scores are reported for selected training runs, and observations are made about the resulting plots indicating adversarial incentives lead to more reward hacking.

**Strengths:**

1. The paper operationalises and studies scalable oversight, which is an interesting research challenge.
2. Caveats and limitations are listed along with empirical observations.

**Weaknesses:**

1. **The paper is extremely poorly written and hard to understand.**

Most statements are verbose yet imprecise. It almost seems like a ramble of various disconnected points with no logical flow in many places. I will provide concrete examples. First the introduction. The points in the ‘classification’ are unclear, and it's also unclear what implication they have for the paper. Is this a novel classification contributed by the paper? More examples: Intro L50: ‘a method such as Critiques’ — it is unclear what is being referred to, which makes this whole point hard to understand. What do you mean by overseer mistakes can be harder to learn?

Second, related work barely talks about related papers and the marginal contribution in context of them, but rather is an extended introduction or background section. I would recommend shifting the final 2 paragraphs to the introduction section as ‘contributions’, for example. However, the second bullet point is hard to understand too.

The oversight protocols are hard to understand as described, and don’t provide any reference. Did this paper come up with these, or are they used in prior literature? Similarly with the overseers. How did you make the design choices for each of them?

2. **Most claims seem like post-hoc analysis** of experiments running the three different oversight protocols across the different overseers. No clear hypotheses are stated beforehand, and thus it is difficult to judge whether the experiments designed make sense to study them, and whether the conclusions drawn are free from confounders.

3. It is not clear what the contributions of this paper are as it does not properly distinguish prior work. Is the CodeNames task first studied in this paper? What about the overseer policies and the oversight methods? Which of the conclusions drawn from the experiments are novel?

**Questions:**

Please answer the questions raised in the Weaknesses section and also re-organize the paper so it's easier to judge the contributions, hypotheses, motivating the experimental design based on the hypotheses, and finally the strength of evidence in support of the hypotheses.

---

### Note · Authors · 2024-11-27

**Comment:**

After careful consideration of the reviewer feedback, we have decided to withdraw this submission to allow time for addressing the identified limitations and strengthening the work. We appreciate the reviewers' detailed comments which will help improve future versions of this research.

**Withdrawal Confirmation:**

I have read and agree with the venue's withdrawal policy on behalf of myself and my co-authors.